# Integrated Assessment of Survival, Movement, and Reproduction in Migratory Birds: A Study on Evaluating Reinforcement Success

**DOI:** 10.3390/ani14213128

**Published:** 2024-10-30

**Authors:** Guilin Hu, Lijia Wen, Huashan Dou, Yumin Guo

**Affiliations:** 1School of Ecology and Nature Conservation, Beijing Forestry University, Beijing 100083, China; hgl71180206@bjfu.edu.cn (G.H.); wenlj1992@126.com (L.W.); 2Finnish Museum of Natural History, University of Helsinki, 00014 Helsinki, Finland; 3HulunLake National Nature Reserve, Hulunbuir City 021406, China; douhuashan@163.com

**Keywords:** translocation, reinforcement, post-release monitoring, migration, wildlife management

## Abstract

Wildlife conservation faces increasing challenges, prompting the use of reinforcement techniques to bolster declining populations. In this study, the reinforcement success of releasing captive-born non-wild individuals with the post-release outcomes of wild-rescued individuals was compared, focusing on red-crowned cranes *Grus japonensis*, white-naped cranes *Antigone vipio*, and demoiselle cranes *Anthropoides virgo*. By analyzing data collected between 2010 and 2021 through satellite tracking and field observations, this study assesses the survival, movement, and reproduction of both non-wild and wild individuals. The scoring process indicates that timely field observations and a multidimensional assessment of movement patterns enhance evaluation accuracy. The results suggest that wild individuals exhibit higher scores in these aspects, but statistical differences were not significant possibly due to sample size limitations. Notably, non-wild individuals often show residence, nomadic behaviors, or abnormal migration patterns. Field observations underscore the importance of pairing non-wild individuals with wild counterparts to enhance migration success. Recommendations include promoting social integration between non-wild and wild individuals to increase the likelihood of normal migration under wild individuals’ guidance.

## 1. Introduction

Climate change and human activities pose increasing challenges to wildlife survival [1,2]. These challenges have led to population declines and even the extinction of certain species [3,4,5], prompting the growing use of reinforcement in conservation practices. Reinforcement involves intentionally introducing an organism into an existing population of conspecifics [6]. For captive-released individuals, lacking wild survival and migration experience often results in high mortality rates [7], necessitating measures to enhance their survival to effectively bolster wild populations. Moreover, monitoring the reproduction of these released individuals is essential, as it directly influences the demographic dynamics of wild populations in the future.

Advances in tracking technologies have enabled more precise and continuous monitoring [8,9], particularly beneficial for assessing the success of reinforcement efforts in migratory birds such as storks and cranes [10,11]. Tracking provides crucial data on survival, reproduction, and movement patterns, essential indicators for evaluating reinforcement success. Specifically, detailed movement patterns of released individuals can accurately reflect their migratory behavior [11]. Migration plays a crucial role in bird survival, given the heightened mortality risks during these journeys [12], thereby emphasizing its critical role in evaluating the outcomes of reinforcement efforts. Monitoring released individuals also informs the efficacy of pre-release management strategies [13], guiding future reinforcement practices. So, continuous monitoring not only aids in evaluating success but also in optimizing and refining future reinforcement efforts based on monitoring outcomes.

In reinforcement efforts for migratory birds, cranes face varying degrees of survival threats due to their large size and migratory habits, unlike small-bodied and non-migratory species [14,15]. Of the 15 crane species worldwide, 11 are categorized as vulnerable, endangered, or critically endangered on the IUCN Red List, with declining wild populations for 9 species. Therefore, the translocation for cranes is definitely necessary and urgent.

Currently, translocation, including reinforcement and reintroduction, has been carried out for the Siberian crane *Leucogeranus leucogeranus*, the red-crowned crane *Grus japonensis*, the white-naped crane *Antigone vipio*, and the sarus crane *G. antigone* [16,17,18,19]. The objectives of translocations vary among crane species, for instance, reintroductions of the whooping crane *G. americana* and sarus crane aim to establish new populations [17,20], while for others, the goal is to augment existing wild populations [16]. These differing objectives influence the assessment criteria used for each species. Studies evaluating the success of reinforcement for species like the Siberian crane and red-crowned crane often consider factors such as survival and reproduction [18,19], whereas assessments for the whooping crane also incorporate modeling to predict future population dynamics [20,21,22].

Despite ongoing efforts, current assessments of crane reinforcement or reintroduction often focus on one aspect (survival [19,20,21], reproduction [18,22]), thus lacking comprehensive evaluations of survival, reproduction, and movement. This fragmented approach may lead to an incomplete understanding of post-release outcomes. In this study, the survival, movement, and reproduction were monitored of 18 cranes (13 born in captivity involved in reinforcement efforts and 5 wild cranes) using satellite tracking data, supplemented by field observations. These data enabled a thorough assessment of post-release conditions. By employing this evaluation method, the aim of this study was to provide a more comprehensive reference for improving the reinforcement success in migratory birds.

## 2. Materials and Methods

### 2.1. Banding and Tracking

In this study, 5 wild individuals rescued from their natural habitat and released following recovery, and 13 non-wild individuals born in captivity, released into the wild upon reaching maturity (Table 1) were included. Prior to release, all individuals were equipped with plastic color rings and GPS/BDS-GSM satellite transmitters. The equipment included the following: (1) color ring on the right leg; (2) leg-mounted satellite transmitters on the right leg with a color ring on the left leg; (3) color ring on the right leg with back-mounted satellite transmitters. The satellite transmitters, powered by lithium batteries and solar panels, weighed approximately 0.3–0.7% of the crane’s body weight, less than 4% in total, ensuring no interference with their normal activities [23]. Positional data were transmitted hourly via GSM/BDS, categorized into five accuracy levels: Level A (within 5 m), Level B (within 10 m), Level C (within 20 m), Level D (within 100 m), and invalid. For clarity in subsequent discussions, individuals were identified by abbreviations combining their species’ English names with the numbers of their respective color rings or tracking transmitters (Table 1).

### 2.2. Release into the Wild

Between 2010 and 2021, a total of 18 individuals were released at various times and locations within nature reserves. Specifically, releases took place at the following reserves: Da Zhan River Wetland National Nature Reserve in Heilongjiang Province (referred to as DNR), Longfeng Lake Provincial Nature Reserve in Jilin Province (referred to as LNR), Hulun Lake National Nature Reserve in Inner Mongolia (referred to as HNR), Ar Horqin National Nature Reserve in Inner Mongolia (referred to as ANR), Yellow River Delta National Nature Reserve in Shandong Province (referred to as YNR), and Poyang Lake National Nature Reserve in Jiangxi Province (referred to as PNR) (Figure 1, Table 1).

### 2.3. Data Analysis

Among 18 individuals, 16 individuals had valid tracking data, excluding R1K7 (lost signal immediately after release) and W283 (not equipped with satellite transmitters), which relied solely on field observation data. After filtering out data deemed invalid, subsequent analysis utilized “longitude”, “latitude”, and “time” of 152,515 valid location records (Table 1). 

Survival, movement, and reproduction scores were individually assessed for both non-wild and wild individuals to evaluate reinforcement success. A higher total score indicated better release effectiveness, incorporating the behavioral differences between non-wild and wild individuals. The specific scoring criteria were as follows:

Survival score evaluated whether an individual survived for at least 1 year. Given the higher mortality rates during the initial stage in the wild, we focused on survival within the first year after release to reflect their viability. If an individual’s survival time equaled or exceeded 1 year (as recorded by tracking or field observation data, at least 365 days post-release), they received 1 point. For individuals tracked or observed for less than 1 year, their survival score was calculated based on the proportion of days survived relative to the 365-day period. If an individual died within the first year, they received a death score of −1 point. If the cause of death could not be determined, they received 0 points. The survival score is the sum of the basic survival score and any death score assigned. Determination of death was based on extended periods of no movement or displacement, confirmed through field investigations.

Movement score assessed the migratory capabilities of individuals’ post-release. With the exception of two individuals (W283 and R1K7) lacking tracking data, movement trajectories of others were analyzed using the net squared displacement models (NSD models) and further scored based on actual trajectory. The “MigrateR” package in R software (version 4.2.3) facilitated NSD model application to classify movement into five categories: migrant, mixmig (mixed migration), nomad, resident, and disperser. To prevent misclassification of short-distance movements as migration, the initial delta value (δ) parameter was set using the following species-specific shortest migration distances: 745.8 km for the red-crowned crane [24], 1100 km for the white-naped crane (unpublished data), and 2200 km for the demoiselle crane [25]. Scores of “1”, “0.6”, “0.3”, and “0” were assigned to migrant or mixmig (migration and mixed migration are both considered as individual acquisition of migratory ability), disperser, nomad, and resident patterns, respectively, based on likelihood of migratory ability acquisition. These scores were used as their model scores. Additionally, due to potential abnormal migration behaviors in released individuals (e.g., short distances or incorrect directions), ArcGIS refined the scoring in terms of trajectory. Scores ranged from “0” for no migration to “1” for correct migration to both wintering and breeding sites. The movement score was determined as the average of model scores and trajectory scores. In cases involving multiple years of tracking, the highest score for each year was utilized as the model or trajectory score, as higher scores indicate a closer proximity to the migratory state.

Reproduction score reflected integration into the population and contribution to growth. Field observations determined scores as ≥0.5 for pair bonding and 1 for participating in reproduction. Scores ≥ 0 indicated no records of pair bonding. Due to variations in habitat characteristics and the differing challenges in distinguishing pair bond status, the duration of each field observation may vary accordingly. When an individual was in a stable habitat, such as a breeding ground, wintering ground, or stopover site, researchers used drones or telephoto SLR cameras to survey the area based on real-time satellite tracking data. The target individual can be identified by clear leg ring numbers or distinct physical traits. Participating in reproduction was indicated by behaviors such as brooding chicks, incubating eggs, or mating. Additionally, if the target individual was consistently observed with another individual, this behavior can be classified as pair bonding.

Statistical analysis using the “wilcox.test” function in R software was employed to assess differences in survival, movement, reproduction, and total scores between wild and non-wild individuals. Significance was determined at a threshold of *p* < 0.05.

## 3. Results

### 3.1. Survival Score

Among the five wild individuals, four individuals (WA18, R15, RS05, WS02) survived at least 1 year, while the survival status of one individual (WN59) could not be determined as its signal disappeared (Table 2). Among the 13 non-wild individuals, 6 individuals (W283, WS06, RA41, D004, R1K7, R200) survived at least 1 year, achieving a full survival score (Table 2). For the remaining seven individuals, five lost their signals, making it impossible to ascertain their survival status (D001, D066, R2K6, R198, R199). Additionally, two individuals showed no activity and exhibited temperature fluctuations with environmental changes; however, without field inspection, it was inconclusive whether they had died or if their tracking transmitters dropped (D020, R3K1).

The mean survival score among wild individuals was 0.90 (*n* = 5), which was slightly higher than that among non-wild individuals at 0.70 (*n* = 13). However, statistical analysis indicated that this difference was not significant (*p* = 0.23 > 0.05, Figure 2).

### 3.2. Movement Score

Except for two individuals lacking tracking data (R1K7, W283), the modeling results (Figure 3) show the movement patterns of five wild individuals and elven non-wild individuals in this study. Among the wild individuals, three (R15, WA18, WS02) were classified as migratory for at least one year, one (WN59) as a disperser, and one (RS05) as a nomad. For the non-wild individuals, three (D001, D004, WS06) were classified as migratory or exhibiting mixed migration for at least one year, while eight were categorized as non-migratory (D020, D066, R198, R199, R200, RA41, R2K6, R3K1).

Field observations confirmed that W283 continuously appeared at the wintering site in Japan and the release site DNR (also the regular breeding site) over multiple years, indicating a migratory movement pattern (Table 2). When considering their trajectories, it was observed that among the wild individuals, four migrated in expected directions (R15, WA18, WN59, WS02), while RS05 moved within a limited range before its signal was lost (Figure 4). Among the non-wild individuals, except for R200 and RA41, the remaining individuals undertook relatively long-distance movements (Figure 5). Eight individuals (D001, D004, D020, D066, R2K6, R198, R199, WS06) exhibited movement directions consistent with those of wild individuals. Within this group, movement patterns for two individuals changed; R2K6 shifted from abnormal east–west movement to normal north–south movement (second year in Figure 5), and WS06 transitioned from non-migratory to migratory behavior (fourth year in Figure 5).

Furthermore, the migration of these eight individuals was incomplete (Figure 5). Two reached the summering site but did not return to the wintering site (D001, D004), one reached the wintering site but did not return to the summering site (R2K6), and one only reached a stopover site (WS06). Additionally, four individuals did not reach their regular distribution areas (D020, D066, R198, R199).

In the movement score analysis (Table 2), three out of five wild individuals achieved a full score (WA18, R15, WS02), while only one non-wild individual (W283) achieved a full score among the thirteen individuals.

The mean movement score among wild individuals was 0.75 (*n* = 5), which was slightly higher than that among non-wild individuals at 0.46 (*n* = 13), though statistical analysis did not reveal a significant difference (*p* = 0.15 > 0.05, Figure 2).

### 3.3. Reproduction Score

The field observations reveal that among the wild individuals (Table 2), one individual achieved a full reproduction score by producing offspring (RS05, Appendix A), and another individual formed a pair bond (R15, Appendix A). Among the non-wild individuals, two achieved a full reproduction score by producing offspring (W283, Appendix A; RA41, Appendix A), and two individuals formed pair bonds (WS06, Appendix A; R200, Appendix A). Notably, RA41 paired with other non-wild individuals near the release site, while the remaining three individuals paired with wild individuals, with R200 pairing with a rescued wild individual.

The mean reproduction score for wild individuals (0.30, *n* = 5) was slightly higher than that of non-wild individuals (0.27, *n* = 13), but the difference was not statistically significant (*p* = 0.95 > 0.05, Figure 2).

### 3.4. Total Score

In the total scoring (Table 2, Figure 6), none of the five wild individuals reached a full score, whereas among the thirteen non-wild individuals, one individual achieved a full score (W283).

The mean total score for wild individuals (1.95, *n* = 5) was higher than that of non-wild individuals (1.43, *n* = 13) (Figure 6), although the difference was not statistically significant (*p* = 0.14 > 0.05, Figure 2).

## 4. Discussion

### 4.1. The Integrated Assessment of Survival, Movement and Reproduction

This study combines GPS tracking and observational data to assess the effects of reinforcement after releasing captive-born non-wild cranes, focusing on survival, movement, and reproduction. The primary goal is to increase the number of wild individuals. Therefore, if released individuals survive but do not integrate into the wild population, the objective is not met. For example, R200 and RA41 paired with injured wild cranes and other released non-wild individuals without joining the wild population, a scenario also observed in the red-crowned cranes released in Yancheng, Jiangsu [19]. Considering both survival and movement provides a clearer indication of the effectiveness of reinforcement. Additionally, some non-wild offspring have successfully integrated into the wild population [26], so including reproduction in the evaluation enhances the assessment’s comprehensiveness.

However, the evaluation of reinforcement still has limitations in itself. An important concern in the scoring process is the incompleteness and uncertainty of individual information, which may lead to scores not fully reflecting the actual circumstances. Information incompleteness primarily arises from the loss of satellite transmitter signals, thereby hindering comprehensive monitoring of released individuals’ activities. In assessing survival status, individuals who have neither been confirmed deceased nor received a full score are those whose satellite signals were lost within one year (e.g., D001, R200, D066, R2K6, R1K7, WN59). This loss of signal introduces uncertainty into the future behavioral dynamics of these individuals. For instance, despite the potential for these individuals to have formed pair bonds or even reproduced, data limitations restrict assigning anything other than a zero score in reproduction assessments. Consequently, actual reinforcement success may exceed that indicated by current scoring methods.

Scientific uncertainty, an ongoing challenge in this field [20], underscores the importance of factoring uncertainty into reinforcement success evaluations. For instance, when estimating population persistence, biases stemming from environmental stochasticity, individual variability, and parameter uncertainty must be considered [27]. Addressing scientific uncertainty necessitates the close integration of tracking and field observation data by managers. This integration aims to bolster monitoring efforts and mitigate the influence of uncertainty on assessment outcomes. 

### 4.2. The Difference Between Wild and Non-Wild Individuals After Release

Our research results show that the overall score of non-wild individuals is lower than that of wild individuals, but there is no significant difference. In terms of survival, there is no significant difference between wild individuals and non-wild individuals. In the study of whooping cranes, it is also confirmed that the survival rates of wild individuals and captive-bred non-wild individuals are similar [20]. They all face the mortality threats of collisions with power lines [28], entanglement in fences [29], and predation [21] in the field. Moreover, wild individuals that migrate encounter heightened survival risks due to the complexities of the migratory environment, resulting in a higher mortality rate [12]. Conversely, non-wild individuals under human care may exhibit higher survival rates than healthy wild counterparts [30]. However, during the early stages post-release, the lack of field survival experience in non-wild individuals could lead to elevated mortality rates, akin to the increased mortality observed in younger wild individuals [31]. These comprehensive factors make there be no significant difference in survival between wild individuals and non-wild individuals.

In terms of reproduction scores, there is also no significant difference between wild individuals and non-wild individuals. Whether they are wild or non-wild individuals, we have recorded cases where they paired with other individuals (RA41, R1K7, R200, W283, WS06) and even reproduced successfully (RA41, W283) after being released. This has also been reported in other studies [32,33]. Therefore, we believe that whether an individual is wild or not does not affect its reproductive behavior. However, after the individuals are in the wild, we cannot continuously track all of them to determine their reproductive status, which affects our results to some extent.

In terms of movement scores, though the absence of a statistically significant difference in movement scores between wild and non-wild individuals may be attributed to a small sample size, this does not imply that their movement patterns are inherently similar. None of the seven non-wild individuals with sufficient first-year data migrated normally. Of these, five exhibited abnormal migration patterns (R2K6, R3K1, D001, D004, D066), one remained resident (RA41), and one nomad (WS06). All five wild individuals migrated normally after release; Except for R1K7 with an unknown trajectory, only two out of twelve non-wild individuals (W283 and WS06) exhibited normal migration patterns. Among them, W283 remained in residence for the first two years. In the third year, after pairing with a wild individual, it migrated successfully. WS06 migrated to Tianjin, and our field observations show that its movement was also led by wild individuals. The migration of cranes depends on social learning and experiential learning [34]. Our research results also indirectly confirm this. The movement patterns of non-wild individuals after being released into the wild may be affected by other wild individuals. However, this needs to be further confirmed by research. 

### 4.3. The Characteristics of Non-Wild Individuals After Release

Among non-wild individuals, there exists a variation in total scores despite some receiving full marks for both survival and reproduction, primarily influenced by differences in their movement patterns. Specifically, individuals such as W283 and WS06 migrated normally, whereas others like RA41 and R200 did not migrate and opted to reside or nomad. This difference underscores the influence of pair bonding on the movement of released non-wild individuals, similar to observations in reintroduced whooping cranes. For instance, a non-migratory male whooping crane joined a migratory female after forming a pair bond, altering its movement behavior [33]. Consequently, intensive monitoring of sexually mature released non-wild individuals is crucial as they may change their movement patterns during this critical stage.

While no field evidence suggests mortality among individuals in terms of survival scores, satellite tracking data indicate potential mortality for certain individuals (e.g., D020, R3K1). Previous studies have shown that predation, collisions with power lines, diseases, and gunshot wounds are major causes of mortality among reintroduced whooping cranes [21]. Therefore, enhancing monitoring efforts is essential to ascertain the causes of mortality among released cranes, this helps to further understand the factors that threaten the survival of species.

Furthermore, in movement scoring, while the model may assign full scores, some individuals (WS06, D004, D001) did not achieve full trajectory scores. Their migratory patterns differ somewhat from those of wild individuals, necessitating a more detailed analysis using both the NSD models and movement trajectories. Notably, individuals like R2K6 and R3K1 exhibit abnormal migration directions in trajectory scoring. In such cases, intervention measures should ensure adequate food sources and suitable wintering habitats to prevent mortality from harsh environmental conditions [10].

## 5. Conclusions

The ‘Survival-Movement-Reproduction’ scoring method employed in this study provides a comprehensive assessment of reinforcement success and facilitates interpretation of factors contributing to variations in success rates among released individuals. Based on these evaluation outcomes, two recommendations are proposed for enhancing future reinforcement efforts.

Firstly, to refine the assessment of reinforcement success, the following is proposed: (1) When tracking data show abnormal activity, prompt on-site verification is needed to confirm death or tracker detachment, ensuring more accurate survival scores. (2) Intensify the monitoring of released non-wild individuals nearing sexual maturity and promptly verify their pairing status in response to changes in original movement patterns. (3) When scoring movement patterns, fully incorporate the variability observed in released non-wild individuals and integrate the NSD models with trajectory analyses for scoring accuracy.

Secondly, while the use of ultralight aircraft to guide inexperienced non-wild individuals along migratory routes has proven effective in the reintroduction of whooping cranes [13], this approach is both costly and challenging to implement on a broader scale. Consequently, we propose enhancing the probability of successful migration for non-wild individuals by facilitating their pairing or integration with existing groups. Specifically, during the breeding or summering periods, it is advisable to release non-wild individuals approaching sexual maturity in proximity to wild subadult flocks. Additionally, during the overwintering phase, these individuals should be released in areas where wild populations are concentrated.

## Figures and Tables

**Figure 1 animals-14-03128-f001:**
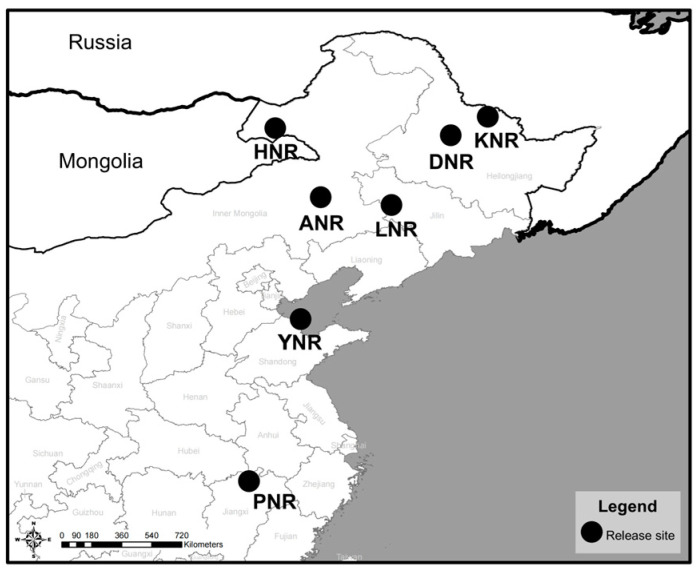
Release site.

**Figure 2 animals-14-03128-f002:**
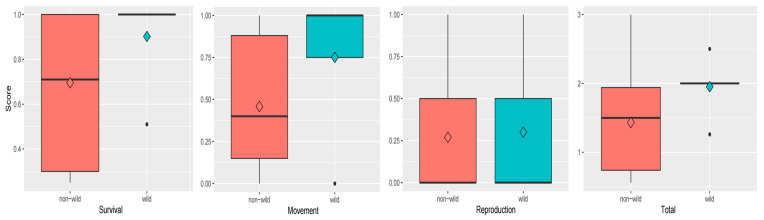
The box plots for the scores of survival, movement, reproduction, and total scores. The diamonds in the box plot represents the mean value for each one. The black dots represent outliers. The black horizontal lines represent the median.

**Figure 3 animals-14-03128-f003:**
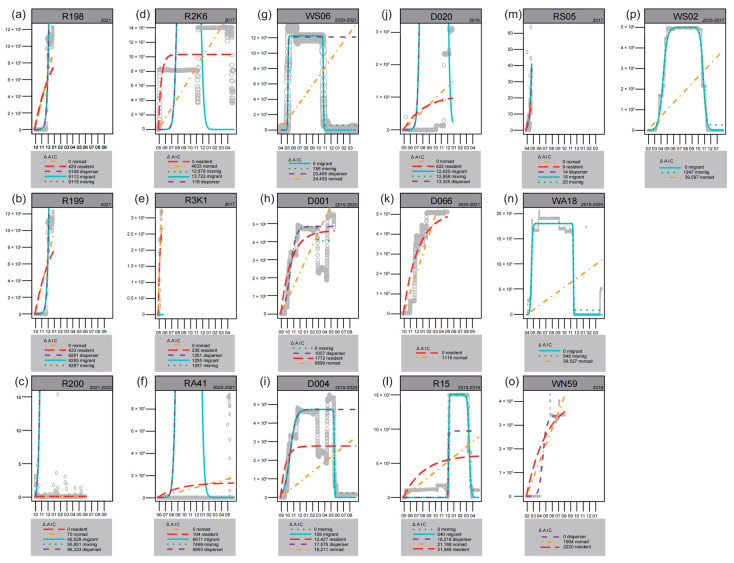
The NSD models classified the movement patterns into different categories for individuals with enough tracking data (*n* = 16). The y axis represents the NSD values, and the x axis represents the months. The grey circles represent actual values, the colored lines represent fitted values. In the box of AIC values, the movement pattern with a value of 0 has the best fitting effect. In the model scoring, the scoring results are based on the closest year to migration, so only the representative NSD modeling results of a specific year are presented here. (**a**–**k**) The modeling results for non-wild individuals. (**l**–**p**) The modeling results for wild individuals.

**Figure 4 animals-14-03128-f004:**
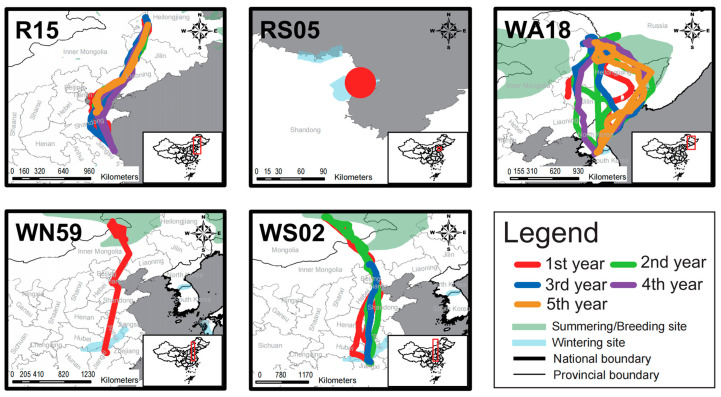
The movement trajectories of wild individuals (*n* = 5).

**Figure 5 animals-14-03128-f005:**
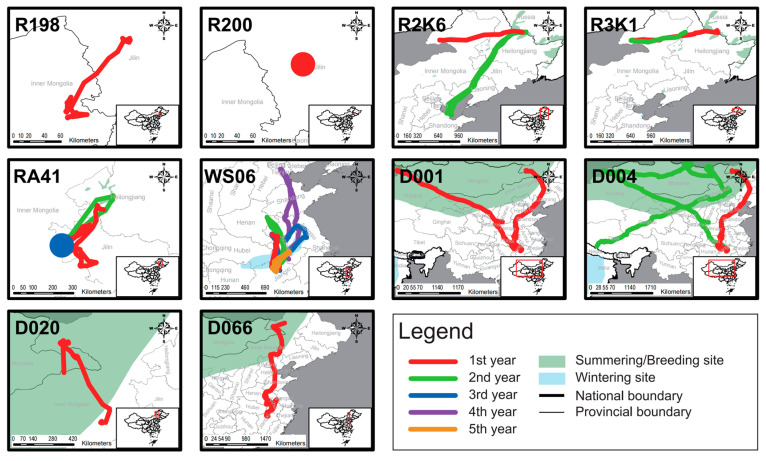
The movement trajectories of non-wild individuals (*n* = 10). W283 and R1K7 lack tracking data, so there is no trajectory plot for them. The movement trajectory of R198 represents both R199 and R198 as they have similar movement trajectories. R200 and RA41 did not undergo significant movement in a particular year, this is represented by a dot.

**Figure 6 animals-14-03128-f006:**
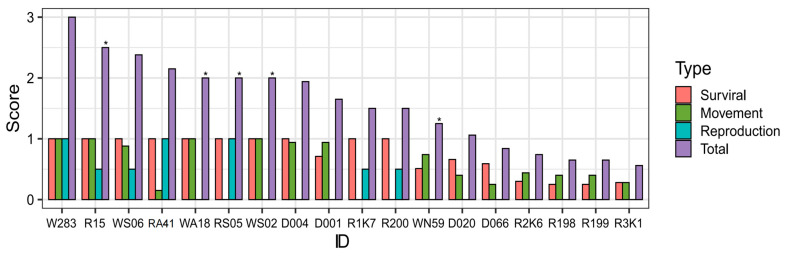
The scoring results for survival, movement, reproduction, and total scores (*n* = 18). Above the bar of total score, “*” represents the wild individuals.

**Table 1 animals-14-03128-t001:** Research individual’s basic information.

Species	ID	Release Age	Release Time	Release Area	Tracked Time	Data Volume
RCC	R198	1−	15 Sep. 2021	LNR ^N^	15 Sep. 2021–13 Dec. 2021	2131
RCC	R199	1−	15 Sep. 2021	LNR ^N^	15 Sep. 2021–13 Dec. 2021	2143
RCC	R200 ^F^	1+	15 Sep. 2021	LNR ^N^	15 Sep. 2021–19 May 2022	3692
RCC	R2K6	1+	16 May 2016	HNR ^S^	17 May 2016–12 Jun. 2016	154
		2+	24 Apr. 2017	HNR ^S^	24 Apr. 2017–17 May 2017	559
RCC	R3K1	1+	16 May 2016	HNR ^S^	16 May 2016–20 Jun. 2016	268
		2+	24 Apr. 2017	HNR ^S^	24 Apr. 2017–12 May 2017	437
RCC	R1K7 ^F^	NA	16 May 2016	HNR ^S^	-	-
RCC	RA41 ^F^	2+	10 May 2018	ANR ^N^	10 May 2018–10 Mar. 2023	26,919
WNC	WS06 ^F^	2+	20 Jan. 2017	PNR ^W^	20 Jan. 2017–3 Nov. 2021	19,384
WNC	W283 ^F^	1+	28 Jun. 2010	DNR ^S^	-	-
DC	D001	NA	28 Aug. 2019	HNR ^S^	28 Aug. 2019–14 May 2020	5191
DC	D004	NA	28 Aug. 2019	HNR ^S^	28 Aug. 2019–14 Oct. 2020	9715
DC	D020	NA	17 Apr. 2019	HNR ^S^	17 Apr. 2019–12 Dec. 2019	2213
DC	D066	NA	8 Sep. 2020	HNR ^S^	8 Sep. 2020–10 Apr. 2021	5108
RCC	* R15 ^F^	NA	15 Apr. 2018	LNR ^N^	15 Apr. 2018–16 Apr. 2023	19,310
RCC	* RS05 ^F^	4+	14 Mar. 2017	YNR ^W^	14 Mar. 2017–11 Apr. 2017	243
WNC	* WA18	1+	18 Mar. 2018	LNR ^N^	18 Mar. 2018–17 Mar. 2023	34,610
WNC	* WN59	1+	20 Jan. 2016	PNR ^W^	20 Jan. 2016–24 Jul. 2016	808
WNC	* WS02	1+	20 Jan. 2016	PNR ^W^	20 Jan. 2016–4 Apr. 2018	19,630

In the species column, RCC represents red-crowned crane, WNC represents white-naped crane, DC represents demoiselle crane. In the ID column, “^F^” represents field observation, “*” represents wild individuals. In the Age column, NA represents individuals lacking age records before release. In the Release Area column, “^S^” represents the summering site, while “^W^” represents the wintering site, and “^N^” does not represent either a summering site or a wintering site.

**Table 2 animals-14-03128-t002:** Score of survival, movement, and reproduction (*n* = 18, ranked in descending order of scores).

ID	Survival			Movement			Reproduction	Total
	Not Die	Die	Total	Model	Trajectory	Average		
W283	1.00	0.00	1.00	NA	1.00	1.00	1.00	3.00
R15 *	1.00	0.00	1.00	1.00	1.00	1.00	≥0.50	≥2.50
WS06	1.00	0.00	1.00	1.00	0.75	0.88	≥0.50	≥2.38
RA41	1.00	0.00	1.00	0.30	0.00	0.15	1.00	2.15
WA18 *	1.00	0.00	1.00	1.00	1.00	1.00	≥0.00	≥2.00
RS05 *	1.00	0.00	1.00	0.00	0.00	≥0.00	1.00	≥2.00
WS02 *	1.00	0.00	1.00	1.00	1.00	1.00	≥0.00	≥2.00
D004	1.00	0.00	1.00	1.00	0.875	0.94	≥0.00	≥1.94
D001	0.71	0.00	0.71	1.00	0.875	0.94	≥0.00	≥1.65
R1K7	1.00	0.00	1.00	NA	NA	≥0.00	≥0.50	≥1.50
R200	1.00	0.00	1.00	0.00	0.00	0.00	≥0.50	≥1.50
WN59 *	0.51	0.00	0.51	0.60	0.875	0.75	≥0.00	≥1.26
D020	0.66	0.00	0.66	0.30	0.50	0.40	≥0.00	≥1.06
D066	0.59	0.00	0.59	0.00	0.25	0.13	≥0.00	≥0.84
R2K6	0.52	0.00	0.30	0.00	0.875	0.44	≥0.00	≥0.74
	0.07	0.00						
R198	0.25	0.00	0.25	0.30	0.50	0.40	≥0.00	≥0.65
R199	0.25	0.00	0.25	0.30	0.50	0.40	≥0.00	≥0.65
R3K1	0.52	0.00	0.28	0.30	0.25	0.28	≥0.00	≥0.56

In the ID column, “*” represents wild individuals. Individuals with “NA” in the model score column only have observational data available. RS05 and R1K7 have known breeding sites but unknown wintering sites, indicating the possibility of migration; therefore, their movement score is “≥0”. Individuals with known mating pairs have the possibility of reproduction; hence, their reproduction score is “≥0.5”. For other individuals lacking observational data, there is still a possibility of mating, so their reproduction score is “≥0”.

## Data Availability

Due to the sensitive nature of the data involving the protection of threatened species, the location-specific data from this study are not publicly available.

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
