# Peer review of "Integrated Assessment of Survival, Movement, and Reproduction in Migratory Birds: A Study on Evaluating Reinforcement Success"

_animals, 2024, doi:10.3390/ani14213128_

Round 1

Reviewer 1 Report

Comments and Suggestions for Authors

see attached.

Author Response

Comments 1:

L12-13: In my opinion, Simple Summary, Abstract and text are independent parts of an article, therefore, any animal name first appeared should give a Latin name. But if not, you may change as “focusing on three crane species”

Response 1:

I had added Latin names in L12-13.

Comments 2:

L43-44: Current factors such as climate change and human activities pose increasing challenges to wildlife survival, including habitat loss and fragmentation [1,2].
Revised as:
Climate change and human activities pose increasing challenges to wildlife survival [1,2].
What’s about current factors? In fact, you do not study habitat loss and fragmentation in this paper.
Please also use two references, one is for Climate change and another for human activities, to pose increasing challenges to wildlife survival.
Cited ref. 1, 2 were bad references, do not support your view “Climate change and human activities pose increasing challenges to wildlife survival”.

Response 2:

Thank you for pointing this out. It is indeed true that habitat loss and fragmentation had no relationship with this study. Therefore, in L43 - 44, I have changed the text to "Climate change and human activities pose increasing challenges to wildlife survival". Besides, I have changed ref. 1, 2 accordingly, the ref paper titles are: 
ref.1 : Winter mortality of a passerine bird increases following hotter summers and during winters with higher maximum temperatures.
ref.2 : The impact of hunting on tropical mammal and bird populations.

Comments 3:

L447: It should be Ma, Z. 

Response 3:

I have corrected ref. 23 "Ma, Z. J." into "Ma, Z" in L449.

Lastly, the revision is in the word document. Please see the attachment. Thank you again.

Reviewer 2 Report

Comments and Suggestions for Authors

The authors should amend the paper and consider the comments detailed in the attached pdf.

Main critical points:

1) The authors need to better define wild and non-wild birds. Were non-wild individuals born in captivity, or kept in captivity for a long period ? This should be clarified both in Summary/Abstract, and in the main text.

2) Necessary information about the species of each individual needs to be added to Methods.

3) Further check is advisable for relevant and updated literature not yet included in References (e.g. doi.org/10.1016/j.biocon.2023.110185, DOI: 10.3354/esr01151, doi.org/10.5751/JFO-00412-950106)

4) The number of birds given at several other points throughout the paper is confusing. Rows 126-128 declare that only 16 birds had valid tracking data, and the remaining 2 individuals only had field observations, but then it seems that wrong numbers are given when various results are reported. Please check the numbers of individuals you give while discussing each result

5) Chapter “5.Conclusions” is convincing when the authors propose how to "refine the assessment of reinforcement success".  On the other hand, when they discuss "enhance reinforcement success", no realistic solutions are proposed. It sounds hardly feasible to "Mitigate mortality, Accurately determine the causes of death, Implement appropriate interventions to ensure access to food sources and suitable habitats, Improve migration success probability" for birds released in the wild. These last proposals should therefore be omitted.

Other minor changes, suggested in the attached pdf, aim to a clearer and more concise style.

Comments on the Quality of English Language

I feel that the paper should be revised for clearer and more concise style.

Author Response

Thank you very much for taking the time to review this manuscript. Please find the detailed responses below and the corresponding revisions highlighted changes in the re-submitted files.

Comments 1:

The authors need to better define wild and non-wild birds. Were non-wild individuals born in captivity, or kept in captivity for a long period ? This should be clarified both in Summary/Abstract, and in the main text.

Response 1:

Non-wild individuals were born in captivity, with out any experience in the wild.

I have added 'captive-born’ or 'born in captivity" in four parts as followed:

  1. Simple Summary: L15.
  2. Abstract: L32
  3. Introduction: L98.
  4. Materials and Methods: L110.

Comments 2:

Necessary information about the species of each individual needs to be added to Methods.

Response 2:

In table 1 (L123-124), I added a column named "Species" in the first colume.

Comments 3:

Further check is advisable for relevant and updated literature not yet included in References (e.g. doi.org/10.1016/j.biocon.2023.110185, DOI: 10.3354/esr01151, doi.org/10.5751/JFO-00412-950106)

Response 3:

I added references about reintroduction or reinforcement of crane to further prove some of my discussion, the number of them is [29], [32], [34].

Comments 4:

The number of birds given at several other points throughout the paper is confusing. Rows 126-128 declare that only 16 birds had valid tracking data, and the remaining 2 individuals only had field observations, but then it seems that wrong numbers are given when various results are reported. Please check the numbers of individuals you give while discussing each result.

Response 4:

  1. L220-221: Thank you for pointing out the error here. Since two individuals lack GPS data, the number of non-wild individuals used for modeling is indeed 11.
  2. L323-325: Among the 13 non-wild individuals, 11 individuals with tracking data and 1 individual with only observation data can obtain clear movement trajectories. So here the original 13 individuals are changed to 12 At the same time, it is supplemented in the front that ’Except for R1K7 with an unknown trajectory‘.

Comments 5:

Chapter “5.Conclusions” is convincing when the authors propose how to "refine the assessment of reinforcement success".  On the other hand, when they discuss "enhance reinforcement success", no realistic solutions are proposed. It sounds hardly feasible to "Mitigate mortality, Accurately determine the causes of death, Implement appropriate interventions to ensure access to food sources and suitable habitats, Improve migration success probability" for birds released in the wild. These last proposals should therefore be omitted.

Response 5:

  1. I acknowledged the challenge of reducing mortality solely by identifying causes of death. However, prompt on-site verification of abnormal activity detected via GPS data allows us to confirm deaths promptly. So in L449-451, I added a text, ‘When tracking data shows abnormal activity, prompt on-site verification is needed to confirm death or tracker detachment, ensuring more accurate survival scores'  to enhance the accuracy of survival scoring.
  2. I omitted the text on 'implementing interventions for food and habitat access' but added the suggestion about how to enhance the probability of successful migration for non-wild individuals by facilitating their pairing or integration with existing groups in L456-464. Because in this study, two non-wild white-naped cranes (WS06, W283) migrated in the same direction as the wild population after pairing with wild individuals. These findings suggest that releasing captive-born, non-wild cranes near wild populations to improve migration success is a viable strategy.

Additional clarifications

  1. Thank you for highlighting unnecessary words in my manuscript, I have omitted them as advised.
  2. L52-54: In introduction, firstly I made some adjust for the first sentence as the other reviewer advised, and changed ref [1,2] to prove the sentence. I did not change the ref [3-5], because they are classical examples about population decline or extinction. 
  3. L93: The brackets I added show from what aspects the paper specifically evaluates the reinforcement or reintroduction work.
  4. L97-98: I did not change the text 'from 13 non-wild cranes involved in reinforcement efforts and 5 wild cranes' into 'from 13 non-wild and 5 wild cranes involved in reinforcement efforts,', because the reinforcement efforts solely pertain to non-wild cranes. However, I put the specific situation of an individual in parentheses in order to make this sentence look more fluent.
  5. L184-192: I supplemented the details of field observations, including time, place, observed content, etc. And in the ID column of table 1 (L123), I have added using "F" to represent that these individuals have field observation data. In addition, in the Appendix A file, the names of the observers are indicated. 
  6. L214-216:In figure 2, I mistakenly confused the median with the average. The average is now shown with diamonds while the horizontal line represents the median.
  7. L285: After writing about the limitations of this kind of research in the first part, it is indeed beneficial for explaining the specific research results later. Therefore, I changed the title of 4.1. into ' The integrated assessment of survival, movement and reproduction', and after summarizing the evaluation methods, the limitations of the study are explained as before.
